# A GNSS-Based Crowd-Sensing Strategy for Specific Geographical Areas

**DOI:** 10.3390/s20154171

**Published:** 2020-07-27

**Authors:** Chuan-Bi Lin, Ruo-Wei Hung, Chi-Yueh Hsu, Jong-Shin Chen

**Affiliations:** 1Department of Information and Communication Engineering, ChaoYang University of Technology, Taichung 413310, Taiwan; cblin@cyut.edu.tw; 2Department of Computer Science and Information Engineering, Chaoyang University of Technology, Taichung 413310, Taiwan; rwhung@cyut.edu.tw; 3Department of Leisure Services Management, Chaoyang University of Technology, Taichung 413310, Taiwan; cyhsu@cyut.edu.tw

**Keywords:** epidemic management, GNSS/GPS, infectious disease, isolation, social networking service, hot spot

## Abstract

Infectious diseases, such as COVID-19, SARS, MERS, etc., have seriously endangered human safety, economy, and education. During the spread of epidemics, restricting the range of activities of personnel is one of the options for the prevention and treatment of infectious diseases. A global navigation satellite system (GNSS), it can provide accurate coordinates of latitude and longitude to targets with GNSS receivers. However, it is not common to use GNSS coordinates to represent positions in social life. For epidemic management, it is important to know the locations (and addresses) of targets, especially in social life. When there are many targets, it is not easy to efficiently map these coordinates to locations. Therefore, we propose a GNSS-based crowd-sensing strategy for specific geographical areas that can be used to calculate how many targets are in specific geographical areas or whether a target is in a specific area. This strategy is based on the coordinates of latitude and longitude provided by GNSS to find the locations of these coordinates. As simulated data, the data records containing latitude and longitude in a well-known social networking service platform are used. The strategy is also available for mining hot spots or hot areas.

## 1. Introduction

A satellite navigation system with global coverage is called a global navigation satellite system (GNSS). It allows small electronic receivers to find its position in longitude and latitude coordinates. Current international GNSS standards for International Civil Aviation address only two core constellations: the U.S. Global Positioning System (GPS) and the Global Navigation Satellite System (GLONASS) [1,2]. These systems allow small electronic receivers to determine their location with the values of longitude and latitude to high precision using time signals transmitted along a line of sight by radio from satellites. Moreover, GPS receivers released in 2018 that use the L5 band [3] can have much higher accuracy. Many studies, that have used different techniques to improve observation precisions of GNSS positioning [4,5,6], can support this study. In this study, the corresponding locations of the latitude and longitude coordinates provided by a GNSS are evaluated. We also assume there is a large number of targets. Each target has a GNSS receiver that can acquire its current latitude and longitude coordinates and delivers its coordinates to a server. The coordinates of targets can be acquired by a computer manner. Our proposition is to assume that, when spaces of specific areas are large, such as New York City, Wuhan City China, and Taipei City, the number of targets to evaluate their locations, such as the population of a city, is also large.

During the spread of an epidemic, such as COVID-19 [7,8,9], it is an effective option to limit the movement range of people in order to control the epidemic. Determining the locations of targets can help determine whether people are in quarantine. There are two topics in this research. The first item is whether the target is within a certain range, which can be used to determine whether the quarantined person is in the isolation zone. The second item is the number of targets is in a specific area, thereby, it can be used to determine whether too many people are likely to cause infectious diseases. For example, in Taiwan, the four days from 2 April to 5 April 2020, are a consecutive holiday. During this period, there were a large number of tourists. The National Health Command Center (NHCC) in Taiwan [9] estimates that there were 1.5 million visits in 11 scenic spots through the telecommunications operator signal. Since the average stay time of these visits exceeded 15 min, the command center was worried about triggering a cluster infection of COVID-19 and urgently issued a national police call to the 11 scenic spots to call for evacuation, as well as the health management of tourists.

In general, a location with latitude and longitude is termed as a geographical point. There are very many geographical points generated by various social network services (SNSs) of the platforms, such as Facebook, Twitter, Google, and Foursquare. Check-in for some targets is a location-based service. It provides a mechanism to record users who have visited these geographical points. Among these platforms, Facebook is the largest one with regard to the number of users. In Facebook, these points are termed as “places”. In other words, the check-in places are the locations that people actually visited. So, using these points as experimental samples to explore the crowd distribution is reasonable and credible. The Facebook penetration rate in Taiwan is the highest in the world. Here, the number of daily users reached 13 million of approximately 23 million. Accordingly, Taiwan is an appropriate selection as the experimental area, where the main island has an area of 35,808 square kilometers.

In light of above discussions, this study proposes solutions to the following:For real-time coordinates of targets, this strategy can be used to determine the number of targets in the specified area in time and can also be used to determine whether the target is in the specified area.For historical coordinates of targets, this strategy can be used to determine areas in which the targets can easily gather or which locations are the hot spots.

The rest of the paper is organized as follows. In Section 2, the research related to our study is introduced. In Section 3, the system architecture and the problems are described, in which the proposition is formalized. In Section 4, the crowd-sensing strategy is proposed. In Section 5, a demonstration is given by taking the area of Taiwan Island as the experimental area and the Facebook check-in places as the targets and examples. In Section 6, a discussion about the performance is described. Finally, a conclusion is given in Section 7.

## 2. Related Work

Accurate latitude and longitude positioning supports our research to evaluate the corresponding social locations. In [4,5,6], there are different techniques to improve observation precisions of GNSS positioning. In [10], the authors focused on the integrated methodology of GNSS and device-to-device measurements. The simulation and experimental results demonstrated that the integrated methodology outperforms the nonintegrated one. In [11], a two-step approach is studied, namely, computing first the Fisher Information Matrix (FIM) for the channel parameters, and then transforming it into the FIM of the position, rotation, and clock-bias. The analysis demonstrated the advantages of the hybrid positioning in terms of (1) localization accuracy, (2) coverage, (3) precise rotation estimation, and (4) clock-error estimation. In [12], this study presented a wideband/multiband quad-antenna system for 4G/5G/GPS metal-frame mobile phones. The merit of the proposed antenna system is that a quad-antenna system is achieved under the condition of a metal frame and, without using any decoupling structure, the desired bands for 4G/5G/GPS are covered.

The Internet environment has generated a large number of geographical points, such as Facebook check-in places [13,14,15], Google Maps places, Foursquare check-in places, etc. These places in Social Networking Services (SNS) are special kinds of geographical points. However, each SNS point contains not only a geographical coordinate and some contents to introduce this point. Accordingly, there are many studies focused on these points with their contents. In [16], the paper aimed to assess the role that interactive technology can play in enhancing urban governance to meet social needs. In [17], a real-time Google Maps-based arterial traffic information system for urban streets is presented. In [18], the authors proposed the reuse of up-to-date and low-cost place data from social media applications for land use mapping purposes by Foursquare place data. In [19], the study aimed to explore Foursquare mobility networks and investigate the phenomena of clustering venues across the cities. In [20], the study aimed to inform on how scientific researchers could utilize data generated in location-based social networks to attain a deeper understanding of human mobility. In [21], the authors proposed to find the geographical points related to a special folk belief. In [22], the author utilized the Markov Chain model with their proposed activity detection method to predict the activity category of the user’s next check-in location. In [23], an urban tourism check algorithm is proposed. It can find those who are tourists and find out where the people come from and the route of their visit. In [24], the structure analysis of place networks is explored, in which vertices of geographic places, while the links between places are formed, are based on the user’s check-in history. In [25], a new feature fusion-based prediction approach is proposed, based on carefully designed feature extraction methods. If these points can be mapped to social locations, not only coordinates, these applications could have further enhancements. To acquire the social locations of geographical points is also a promising topic, such as in epidemic management. In computational geometry, this is the point-in-polygon (PIP) problem [26,27,28,29,30]. These studies provide methods to evaluate that a point is inside an area or is not-inside an area without error. In [29], a computer-friendly method was proposed for the PIP problem.

In [30], the authors developed a PIP algorithm that can evaluate that a point is inside or is not-inside an area. Based on this PIP algorithm, the method to find all points within a specific geographical area was developed. It first plans a range that can cover the entire specific area. Then, it finds all the points in the planned range. For each point, the PIP algorithm is used to calculate whether this point is in the specified area. Because the PIP method is verified, this method can be applied to evaluate all the points in the specified area without error. However, this method is inefficient when the area is large with a large number of points. The planned range is often much larger than the specific area. This method selects a large number of points that are not in this specific area and confirms their locations by PIP. If the planned range can be smaller, the points that need to be confirmed by PIP will be reduced a lot. In addition, if certain points are known in the area or known outside the area, there is no need to confirm through PIP. In this way, the performance will be improved a lot.

## 3. System Architecture and Problem Description

The architecture of the system assumes that the target has a GNSS receiver. It can convert the received signal into coordinate data and send it to storage, such as databases, according to the transmission mechanism of the wired or wireless network. The target will be represented by a geographic point, referring to a point that has its latitude and longitude coordinate. In geometry, a point with coordinate is termed as a geographical point. Moreover, a geographical area (a specific area) is a polygon. A polygon is composed of edges with geographical points. These edges enclose a measurable interior [21]. Moreover, for a geographical point *p*, *x*(*p*) and *y*(*p*) are used to represent the coordinate of *p*. For a geographical area *A* with *n* points, *a*_0_, *a*_1_, …, and *a_n_* are used to represent these *n* points, where *a*_0_ = *a_n_.*

The architecture of this system can be simplified into a geographic area *A* and a set *P* of geographical points. Because the area of area *A* is large and the size of set *P* is also very large, it is impossible in the field of computers to load all the points of *P* into the variables in the programming language and then calculate whether each point is in the range of *A*. It must be planned to only take points in part of the area in *A* at a time and these points are affordable by computers. The type of this area is chosen as a circle. There are two reasons. The first reason is that some well-known platforms (servers), such as Facebook, also provide geographical points in a circle manner. The other reason is to follow the access method of storages. The storage in general is a database system. To retrieve some records from the storage, it must be obtained through the access mechanism of the database system, such as structured query language (SQL). The circle-area manner can be mapped to the corresponding SQL commands. However, the configuration of the circle manner is not easy to fully cover a geographical area. Here the cellular architecture using a regular hexagonal configuration, instead of the circular configuration, can be applied. The cell architecture is shown in Figure 1, in which Figure 1a provides the layout of a cell and Figure 1b provides the relative positions of cell *c* and its six neighbors. If the radius of the circle is *r*, then the length of the regular hexagonal side is *r*. For convenience, a cell is used to represent a regular hexagon. There are six cells around a cell, termed as the neighbors of this cell. For a cell *c*, cells *c*_N0_, *c*_N1_, …, and *c*_N5_ are used to represent the six neighbors of cell *c*. The relative positions of cell *c* and its six neighbors can be expressed by an oblique coordinate architecture with 60 degrees (i.e., the angle between the *x*-axis and the *y*-axis is 60 degrees). Moreover, for cell *c* with radius *r*, Table 1 provides the coordinates of each vertex. Table 2 provides the coordinates of each neighbor.

Moreover, if expressed by the geographic coordinate system (i.e., use latitude and longitude values to describe a coordinate with cell *c*), the offset coordinates of the six neighbors are shown in Table 2. Notably, the distance *r* is transformed by the Cartesian coordinate. Every point that is expressed in ellipsoidal coordinates can be expressed as rectilinear *x y z* (Cartesian) coordinates. Cartesian coordinates simplify many mathematical calculations. The Cartesian systems of different data are not equivalent. The distance between the points of longitude 121 and latitude 21 to the point of longitude 122 and latitude 21 is about 103 km. The distance between the points of longitude 122 and latitude 21 to the point of longitude 122 and latitude 22 is about 111 km. If it is configured with a radius of 1 km, *r* can be set to about 0.009 (=1/111) near this area.

A cell *c* is a regular hexagon area, defined as (*c*, *r*), where (*x*(*c*), *y*(*c*)) is the geographical coordinate, and *r* is the length of the regular hexagonal side. Moreover, V(*c*) = {*c*_V0_, *c*_V1_, *c*_V2_, *c*_V3_, *c*_V4_, *c*_V5_} is defined as the set of six vertices and NB(*c*) = {*c*_N0_, *c*_N1_, *c*_N2_, *c*_N3_, *c*_N4_, *c*_N5_} is defined as the set of six neighboring cells. Table 1 provides the coordinates of each vertex in *V*(*c*) and Table 2 provides the coordinates of each neighbor in *NB*(*c*).

Hereafter, the word “geographical point” is simply termed as a “point” and the word “geographical area” is simply termed as an “area”. Evaluating whether a point is inside or is not-inside is the PIP problem. It can count the intersections of the polygon with the ray of this point. A ray of a point is starting from this point to any fixed direction. That the number of intersections between the polygon and the ray is odd indicates this point is inside this polygon. Otherwise, it indicates this point is not-inside this polygon. This method can evaluate the locations of points with respect to an area without error. To implement this method, it must first be able to determine whether two lines are intersected. In [29], a computer-friendly method was proposed to do it. Suppose there are two lines. The first line crosses over both point *o*_1_ and point *o*_2_. The other line crosses over both point *o*_3_ and point *o*_4_. First, *ρ*, as shown in (1), can be evaluated. If the value of *ρ* equal 0, these two lines are parallel. Next, *κ*_1_ and *κ*_2_, as, respectively, shown in (2) and (3), can be evaluated. If the value of *κ*_1_ is between 0 and 1, the intersection is between *o*_1_ and *o*_2_. If the value of *κ*_2_ is between 0 and 1, the intersection is between *o*_3_ and *o*_4_. In other words, if edge (*o*_1_, *o*_2_) and edge (*o*_3_, *o*_4_) are intersected, the value of *κ*_1_ is between 0 and 1 and the value of *κ*_2_ is also between 0 and 1.
*ρ* = (*x*(*o*_1_) − *x*(*o*_2_)) · (*y*(*o*_3_) − *y*(*o*_4_)) − (*y*(*o*_1_) − *y*(*o*_2_)) · (*x*(*o*_3_) − *x*(*o*_4_))(1)
*κ*_1_ = ((*x*(*o*_1_) − *x*(*o*_3_)) · (*y*(*o*_3_) − *y*(*o*_4_)) − (*y*(*o*_1_) − *y*(*o*_3_)) · (*x*(*o*_3_) − *x*(*o*_4_)))/*ρ*(2)
*κ*_2_ = − (((*x*(*o*_1_) − *x*(*o*_2_)) · (*y*(*o*_1_) − *y*(*o*_3_)) − (*y*(*o*_1_) − *y*(*o*_2_)) · (*x*(*o*_1_) − *x*(*o*_3_)))/*ρ*(3)

The overlap of an area *A* with a cell *c*, where the space of *A* is far greater than the space of *c*, is also considered. There are two overlapping cases between *A* and *c*. The first case is the space of *A* is overlapped with the space of *c*. Non-first relationship is the second relationship. The first case can be evaluated when all six points are inside *A* and there are no intersection points among the edges of *c* and the edges of *A*.

As shown in Figure 2, an area is composed of a convex polygon with six points, a_0_, a_1_, …, and a_5_. The rays of p_1_, p_2_, and p_3_ within this area, respectively, have 2, 1, and 0 intersection(s). Since the ray of a point has odd intersections within the area, it means that the point is inside the area; otherwise, the point is not-inside the area. Then, both point p_1_ and point p_3_ are not-inside the area and point p_2_ is inside the area. Continually, there are three cells, c_1_, c_2_, and c_3_. Since all vertices of c_1_ are in this area, two vertices of c_2_ are in this area, and no vertices of c_3_ are in this area, both c_1_ and c_2_ overlap with this area and c_3_ does not overlap with this area.

## 4. Crowd-Sensing Strategy

Given an area *A* = {a_0_, a_1_, …, a*_n_*} and a large number of points *P*, where *P* are stored in a database and can be accessed by using SQL commands, this strategy includes two step cell allocation and point acquisition. It can achieve the evaluation all of points in *P* inside area *A* by applying the two steps.

### 4.1. Cell Allocation

The first step applies algorithms EEI, PIA, and CAO to achieve cell allocation. Algorithm 1 provides algorithm EEI. It is an Edge–Edge Intersection algorithm, where inputs are two edges (*o*_1_, *o*_2_) and (*o*_3_, *o*_4_) and the output is a value of 0 or 1. If edge (*o*_1_*, o*_2_) and edge (*o*_3_, *o*_4_) are intersected, it results by returning 1. Otherwise, it results by returning 0. In this algorithm, it first calculates the value of *ρ* according to (1). If the value of *ρ* is 0, the two edges are parallel. If the value of *ρ* is not 0, it calculates the values of *κ*_1_ and *κ*_2_. If both *κ*_2_ and *κ*_1_ are between 0 and 1, it indicates that the two edges have an intersection.
**Algorithm****1** EEI (*o*_1_, *o*_2_, *o*_3_, *o*_4_)1. { 2.  *ρ*: = (*x*(*o*_1_) − *x*(*o*_2_)) · (*y*(*o*_3_) − *y*(*o*_4_)) − (*y*(*o*_1_) − *y*(*o*_2_)) · (*x*(*o*_3_) − *x*(*o*_4_));3.  if *ρ* = 0 then return 0;4.  else5.  {*6.*    *κ*_1_: = ((*x*(*o*_1_) − *x*(*o*_3_)) · (*y*(*o*_3_) − *y*(*o*_4_)) − (*y*(*o*_1_) − *y*(*o*_3_)) · (*x*(*o*_3_) − *x*(*o*_4_)))/*ρ*;*7.*    *κ*_2_: = − (((*x*(*o*_1_) − *x*(*o*_2_)) · (*y*(*o*_1_) − *y*(*o*_3_)) − (*y*(*o*_1_) − *y*(*o*_2_)) · (*x*(*o*_1_) − *x*(*o*_3_)))/*ρ*;*8.*    if (*κ*_1_ ≥ 0 and *κ*_1_ ≤ 1) and (*κ*_2_ ≥ 0 and *κ*_2_ ≤ 1) then return 1;*9.*    else return 0;10.  }11. }

Algorithm 2 provides the Point Inside Area (PIA) algorithm, whose inputs are a point *p* and an area *A* and its output is a value of 0 or 1. If point *p* is inside area *A*, it results by returning value 1. Otherwise, it results by returning value 0. In this algorithm, variables *o*_1_ and *o*_2_ are used to represent an edge of area *A* and variables *o*_3_ and *o*_4_ are used to represent the incremental extension from *p* toward the *x*-axis until it is greater than the coordinate values of x(*o*_1_) and x(*o*_1_), as in line 6 (i.e., x(*o*_4_) is set as (max(x(*o*_1_), x(*o*_2_)) + 1). Variable count is used to record the number of intersections between the extension line of *p* and the edge of *A*. If the number of intersections is odd, it means that *p* is inside *A*, otherwise *p* is not-inside *A*.
**Algorithm 2** PIA (*p*, *A*) {  *count*: = 0;  for *i*: = 0 to *n* − 1 do  {   *o*_1_: = *a_i_*; *o*_2_: = *a_i_*_+1_;   *o*_3_: = *p*; *o*_4_: =(max(x(*o*_1_), x(*o*_2_)) + 1, y(*p*));   if (EEI(*o*_1_, *o*_2_, *o*_3_, *o*_4_) == 1) then *count*: = *count* + 1;  }  if (*count* % 2 == 1) then return 1;  else return 0; }

Algorithm 3 provides the Cell-Area Overlap (CAO) algorithm, whose inputs are a point *p* and an area *A* and its output is a value of 0 or 1. In this algorithm, each vertex of the cell is taken out independently, and is then calculated for whether it is inside *A*. The variable count is used to record how many vertices of cell *c* are inside *A*. If any of the six vertices of cell *c* are inside *A, c* and *A* are overlapping. Otherwise, *c* and *A* are non-overlapping.
**Algorithm 3** CAO (*c*, *A*) {  *count*: = 0;  for *i*: =0 to 5 do  {   *p*:= *c*_v*i*_;   if (PIA(*p*, *A*) == 1) then *count*: = *count* + 1;  }  if (*count* > 0) then return 1;  else return 0; }

Algorithm 4 provides the Cell Allocation (CA) algorithm, whose inputs are an area *A* and a cell *c_s_*. The allocation starts with cell *c_s_*, called the seed cell, and then expands to its neighboring cells NB (*c_s_*). Then, the neighbors of *c_s_* continue to extend the allocation of cells. The allocation is done until area *A* is completely covered by cells. In this algorithm, variable *C* records the allocated cells and variable *P* records the reference cells. In each iteration (in lines 5 to 14), each cell *c_p_* will be selected in sequence from *P*. Then, the neighbors of the *c_p_* are calculated using NB(*c_p_*) (in line 7). For each cell, *c_NB_* of NB(*c_p_*), if cell *c_NB_* is overlapped with *A* and *c_NB_* is not included in *C*, *c_NB_* will be added into *C* (i.e., *C*: = *C*∪{*c_NB_*}). Additionally, *c_NB_* is added to *N* (i.e., *C*: = ∪{*c_NB_*}). At end of the iteration, the value of *N* is assigned to *p* to start a new iteration. Therefore, this algorithm is stopped when *P* is empty (in line 3) and then it returns the allocated cells *C*.
**Algorithm 4** CA (*c_s_*, *A*) {  *P*: = {*c_s_*}, *N*: = Ø, *C*:= {*c_s_*};  while (*P*≠Ø)  {   for each *c_p_*∈ *P* do   {    *NB*:= NB(*c_p_*);    for each *c_NB_*∈ *NB* do    {     if (CAO(*A*, *c_NB_*) == 1 && *c_NB_* ∉ *C* ) then     {      *N*:= *N*∪{*c_NB_*}, *C*:= *C*∪{*c_NB_*};     }   }   *P*: = *N*, *N*: = Ø;  }  return *C*; }

As shown in Figure 3, the irregular area is composed of 2055 points, and the numbers on cells index the order of cell allocation. In this example, *c*_i_ is used to represent the cell, numbered as *i*. Initially, cell *c*_0_ is allocated. In the first iteration, the six neighbors *c*_1_, *c*_2_, …, and *c*_6_ are included in *C*. Herein, *C* contains seven cells, *c*_0_, *c*_1_, *c*_2_, …, and *c*_6_, and *P* contains six cells, *c*_1_, *c*_2_, …, and *c*_6_. In the second iteration, cells *c*_1_, *c*_2_, …, and *c*_6_ will be the reference cells. For instance, the neighbors of *c*_1_ contains cells *c*_7_, *c*_8_, *c*_2_, *c*_0_, *c*_6_, and *c*_9_. However, *c*_2_, *c*_0_, and *c*_6_ are already included in *C*. Only *c*_7_, *c*_8_, *c*_9_ are included in *C*. Continually, in the third iteration, cells *c*_19_ to *c*_32_ are included to *C* and in fourth iteration, cells *c*_33_ to *c*_37_ are included in *C*. In the fifth iteration, cells *c*_33_ to *c*_37_ are the reference cells *P*. Since all of the non-overlapped neighbors of *c*_33_, *c*_34_, …, or *c*_37_ are included in *C*, no cell are included in *C* and *P* is empty (in line 3). Finally, the allocation is finished by returning the allocation *C*.

### 4.2. Point Acquisition

Points in the local database can be obtained with the similar SQL command in (4), where ‘*P*’ is the table that stores all of geographic points, ‘x’ and ‘y’ are the fields for storing latitude and longitude values, and *r* is the radius of the circular area. In addition, the SQRT (v) function is used to calculate the square root of the value *v*, and the POWER (*v*, *n*) function is used to calculate the *n*th power of the value *v*.
 SELECT * FROM ‘*P*‘ WHERE SQRT(POWER(‘x‘ − x(c),2) + POWER(‘y‘ − y(c), 2)) ≤ *r*(4)

Algorithm 5 provides the Cell Inside Area (CIA) algorithm, whose inputs are a cell *c* and an area *A* and its output is a value of 1 or 0 that, respectively, indicates *c* inside *A* or *c* not-inside *A*. In this algorithm, each edge of *c* and each edge of *A* will be calculated to the number of intersections. If the number of intersections is 0, c is inside *A*, otherwise *c* is not inside *A*.

In summary of the above algorithm, the Crowd-Sensing (CS) algorithm is purposed. Algorithm 6 provides the CS algorithm. It is a Crowd-Sensing algorithm, where the input includes a seed of cell *c_s_*, an area *A*, and all of points *P* and its output is a set *P_A_* of all points inside *A*. In this algorithm, it first calculates the cell allocation *C* by applying CA (*c_s_*, *A*), where *C* is the set of cells that can fully cover the space of area *A*. Then, for each cell *c* in *C*, it first extracts the geographic points *Pc* included in cell *c* from the database. There are two cases to deal with *Pc*. The first case based on cell *c* is inside area *A*, and *Pc* is directly included in *P_A_*. The other case is based on cell *c* not-inside *A*. In this case, every point *p* in *Pc* will be evaluated with area *A*. If point *p* is inside area A, then *p* is included to *P_A_*. For example, in Figure 3, because cells *c*_0_, *c*_1, …,_
*c*_6_, *c*_8,_
*c*_9, …,_
*c*_12_ are inside this area, the points in this range are included in *P_A_*. Other cells are not-inside this area, so the points in this range must be evaluated. In particular, our strategy only needs to evaluate the points in the outermost cells. In fact, the range of points to evaluate is very small. In general, if an area is allocated thousands of cells, only hundreds of cells are evaluated.
**Algorithm 5** CIA (*c*, *A*) {  *count*: = 0;  for *i*: =0 to 5 do  {   for *j*: =0 to *n*-1 do   {    *o*_1_: = *c*_V*i*_, *o*_2_: = *c*_V*i*__+1_, *o*_3_: = *a_j_*, *o*_4_: = *a_j_*_+1_;    if (EEI(*o*_1_, *o*_2_, *o*_3_, *o*_4_) == 1 ) *count*: = *count* +1;   }  }  if (*count* == 0) return 1;  else return 0; }

**Algorithm 6** CS (*c_s_*, *A*)
 {  *P_A_*: = Ø;  *C*: = CA (*c_s_*, *A*);  for each c ∈ *C* do  {   *P**_C_*: = (SELECT * FROM ‘*P*‘ WHERE SQRT(POWER(‘x‘-x(*c*),2) + POWER(‘y‘-y(*c*), 2)) <= r)   if (CIA(*c*, *A*) == 1) *P_A_*: = *P_A_* ∪ *P**_C_*;   else   {    for each *p*∈ *P**_C_* do    {     If (PIA(*p*, *A*) == 1) then *P_A_*: = ∪{*p*};    }   }  }  return *P_A_*; }


## 5. Demonstration

The geographical area of experiment is mainly Taiwan Island that is located between 120 degrees to 122 degrees east longitude and 22 degrees to 25 degrees north latitude. It has an area of 35,808 square kilometers with 23.7 million inhabitants. The points are based on the check-in places of Facebook. The points were acquired from Facebook platform in January 2017, for a total of 1,112,188. We used these points as targets and try to find the targets in specific areas. This area currently contains 6 special municipalities, 10 counties and 3 provincial cities. We demonstrate the results according to the 19 subareas. Figure 4 provides the distribution of 19 subareas, the special municipalities are numbered as 1–6, counties are numbered as 7–16, and provincial cities are numbered as 17–19. Each subarea is composed of hundreds to thousands of vertices. For instance, subarea 1 is composed of 2055 vertices. For each subarea, the proposed strategy was applied to evaluate the number of points inside it.

In Section 5.1, we first display the points in each area. In addition, the population density and the space density with points were also considered to find hot areas. In Section 5.2, we then display the points in each spot. The numbers of points in spots are also used to find hot spots. In addition, the distribution of spots are used to display the crowd distribution. In Section 5.3, we present the contributions of this study. In Section 5.3, we discuss the differences between this study and the previous study.

### 5.1. Points in Areas

In Figure 4a, the value in parentheses is the number of points of the subarea. For example, there are 156,928 points in the area numbered as 4. Moreover, there are a total of 887,950 points in these 19 areas, of which subareas 4, 6, 2, 1, 5, and 3 are the hot areas with the most points. Figure 4b provides the point ratio of each subarea to 16 subareas. There are 616,324 (69.41%) points in these six hot areas. Especially, all of the six hot areas are special municipalities. These results show that, through the coordinates, we can accurately calculate the targets in a specific administrative area, rather than only the information of the latitude and longitude coordinates. Then, the population and the space of hot area are taken into consideration.

Table 3 provides the population, space, points of each hot area, population density, and space density. For example, in subarea 1, the number of the population is 2,687,629, the space is 271.8 square kilometers, and the number of points is 89,209. Therefore, its population density is 3.32% (i.e., the value of 89,209/2,687,629 × 100%) and its space density is 328.21 (i.e., the value of 89,209/271.8). Among these six hot areas, subarea 1 especially has a much higher space density than the other five hot areas. For a specific area with positioned targets, the population and the space can be taken into consideration. It can get the proportion of the targets to the number of the population and the spatial proportion of the target object. This result can be used to determine whether the targets are too crowded in this area based on population or space. It can be used as a control for crowds, such as prohibiting more targets from entering this area or evacuating some targets from leaving this area. It can help to keep the number of targets in this area under control for social applications, such as traffic or epidemic control.

Figure 5 provides the distribution of points for subarea 1 and its neighborhood. The range has 93,549 points, where there are 89,029 points inside subarea 1 and 4520 points not-inside subarea 1. Most of the points are gathered in subarea 1. Moreover, the points are concentrated between longitude 121.45 to 121.6 and latitude 24.95 to 25.01. However, there are almost no geographic points between latitude 25.1 and 25.22. Obviously, the distribution of points is very uneven. Then, we continue to calculate the locations of the hot spots to reveal the concentrations.

### 5.2. Points in Spots

The area is divided into small areas, termed as spots, with a range of 0.01 degrees of latitude and longitude, and the area is slightly larger than 1 square kilometer. As shown in Table 4, there are a total of 15,573 spots. There are 15,347 spots whose numbers of points are less 1000. However, there are 12 spots whose numbers of points are more than 4000. Table 5 provides the details of the 12 hot spots, including the locations, the coordinates, and the numbers of points. The 12 hot spots are in subarea 4, subarea 6, subarea 9, and subarea 19. Moreover, subarea 4 contains six hot spots, subarea 6 contains four hot spots, subarea 9 contains one hot spot, and subarea 19 contains one hot spot. Figure 6 shows the distribution of points in hot spot one. There are 6024 points in this range of about one square kilometer. It means when we are in this hot spot, we are easily exposed to these targets (points). The analysis of hot spots helps to understand whether the target is concentrated in a certain small range, and it is also easy to monitor these spots.

The first 10,000 spots, according to the number of geographic points, are taken into consideration. Then, these spots are classified into 10 groups, where each group is represented using brown color. The darker color represents more geographic points in this spot. The layout of the 10,000 spots is shown in Figure 7a. Comparing the relative positions of the 19 subareas in Figure 4a, it shows that hot spots are almost concentrated in the range of subarea 1 to subarea 6. These six areas are six municipalities in which the population densities are more than other administrative areas. Figure 7b shows the distribution of Highway in this area. The distribution of spots is consistent with the distribution of highway. The geographic points of this experiment are the check-in places of Facebook in Taiwan Island. These places are the historical records of Facebook users who have been to these places for a long time. Because Facebook users are very numerous in this experimental area, these points represent almost every point covered by social activities in the area. Therefore, areas with convenient transportation and high urbanization will have more geographic points. We indicated the spot, according to the number of geographic points, as a spot height map. This map, as shown in Figure 7a, coincides with the highway distribution and administrative area distribution of this area. These facts reveal that our research is a crowd-sensing strategy.

### 5.3. Summary

In [29], an error-free method is presented that can calculate whether a point is inside a polygon. For this, we planned the PIA algorithm, which can also evaluate whether a point is within an area or not without error. Therefore, the efficiency of this type of method is determined by how many points are needed to apply the PIA algorithm and not the accuracy. We take an example, shown in Figure 5, to illustrate the difference between our strategy and [30]. In [30], it first plans a rectangle range that can cover this area. The range is marked as a dashed line. It then finds all points in this range. In our experimental environment, there are 95,349 points. So, the number of executing PIA algorithms is 95,349. In our strategy, we planned a set of cells that could cover this area. The range of the cells is similar to the example shown in Figure 3 that is slightly larger than this area. In fact, we planned with a cell of about 1 square kilometer. This area is about 271.8 square kilometers, as shown in Subarea 1 of Table 3. In this example, the number of cells is about 350 and the number of points within these cells is about 90,000. These cells are classified into inside cells and not-inside cells through the CIA algorithm. Most points are inside cells. The points are surely also in this area without confirming locations by performing PIA. Only a few points located in the outermost cells (not-inside cells) need to confirm their locations. In the examples of Section 5.1 and Section 5.2, the points that need to execute PIA rarely exceeds 10%.

The previous study [30] is very time-consuming to find out the points in large-scale areas, especially when the number of candidate points is very large. The main contribution of our study is that the points that PIA needs to confirm are reduced to very few. Therefore, our strategy has the ability to efficiently handle large-scale areas, such as countries and cities. In the computer field, when the amount of data is large, it is not feasible to process all the data at once. It is necessary to transfer to the batch manner. The design of cells is adaptive to this. It is also convenient to be implemented for epidemic prevention management.

## 6. Discussion

Let *A* be a specific area and *P* be the set of points. Our proposed strategy provided a solution to acquire all points in *P* inside *A*. The PIA algorithm can be used to evaluate whether point *p* is inside or not-inside area *A*. The time complexity is O(*n_A_*), where *n_A_* is the number of vertices (or edges) of area *A*. The reason is that it needs to count the number of intersections among the ray of *p* and the *n_A_* edges. The simple method to acquire all of points inside area *A* is to evaluate each point of *P* by PIP. The candidate points, which are points needing the evaluation of PIA, are all of the *n_P_* points. The time complexity is O(*n_A_* × *n_P_*), where *n_P_* is the number of points in set *P*. When *n_P_* is a very large number, it is greatly time-consuming to acquire these points. In [30], an enhanced strategy was proposed. It first evaluates the boundary (i.e., a space of rectangle). This rectangle is fully covered by the space of area *A*. The points within the rectangle are evaluated. In our strategy, we allocate a set of cells to fully cover area *A*. The total space of these cells is slightly larger than the space of area *A*. Then, the cells are divided into inside cells and not-inside cells. Only points within not-inside cells must be evaluated. The not-inside cells are the cells that have intersections with area *A*. In fact, only points near the edges of area *A* are evaluated. The number of candidate points is reduced to a very small number. Therefore, our strategy is efficient to acquire the targets in an area.

## 7. Conclusions

In this study, Taiwan Island and the administrative areas are used as geographic areas, and check-in places on Facebook are used as geographic points to verify the proposed strategy. These are the actual data. Due to the fact that the spot height map, respectively, matches the distribution of administrative areas and matches the distribution of highways, it verifies the practicability of our strategy. This strategy mainly provides two techniques. The first technique is used to calculate whether the geographic points are in a specific area. The second technique is used to calculate the number of points in a specific area. In our demonstration, we first, respectively, analyzed the numbers of points of the 16 administrative areas. Then the populations of the administrative areas and the spaces of the administrative areas were included in the discussion. This provided these results: the number of geographic points per unit area and the ratio of geographic points to the population. This is the category of hot areas.

Because the numbers of geographic points in geographic areas are not enough to represent the distribution of geographic points, we planned a space of about 1 square kilometer as a unit, termed as a spot, and calculated the number of geographic points in each spot. This is a hotspot category. Of course, the size of a spot depends on the actual situation. Based on the above discussion, our strategy is a GNSS-based crowd-sensing strategy for specific areas. This research is very useful in many fields. We used non-real-time and real-time GNSS coordinates for the applications. Non-real-time coordinates (i.e., historical records), can be used to know where hot areas hot spots can develop. It can be deployed in advance for this area to avoid or mitigate future events. For real-time coordinates, it can be used to know where it is becoming a hot area or where it is becoming a hot spot. It can be deployed ahead of schedule to avoid or mitigate ongoing events. The acquisition of coordinates may have privacy constraints. To acquire current location information with user consent may be available. Observing privacy constraints to do more epidemic prevention management is the goal of our future work.

## Figures and Tables

**Figure 1 sensors-20-04171-f001:**
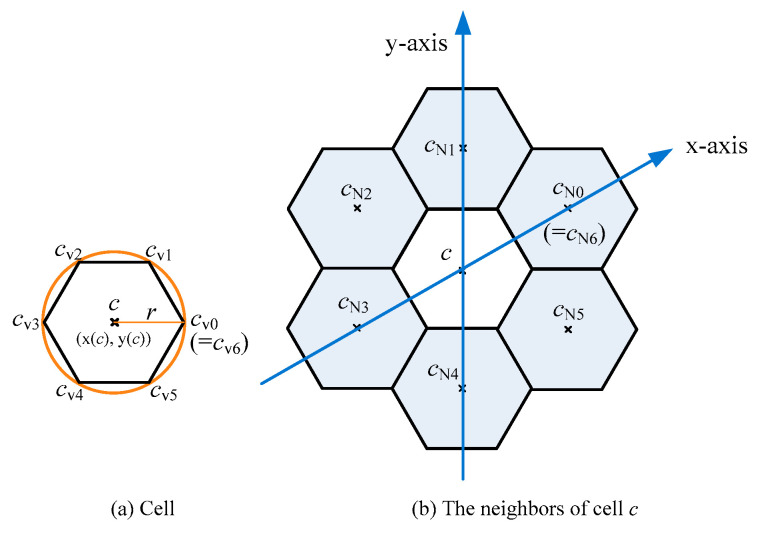
Cell architecture, a cell is a regular hexagon area. (**a**) For cell *c* with radius *r*, *c*_V0_, *c*_V1_, *c*_V2_, *c*_V3_, *c*_V4_, *c*_V5_ are its six vertices. Moreover, *c*_V6_ represents vertex *c*_V0_. (**b**) The neighbors of cell *c* represent as *c*_N0_, *c*_N1_, *c*_N2_, *c*_N3_, *c*_N4_, *c*_N5_. Moreover, cell *c*_N6_ represents cell *c*_N0_.

**Figure 2 sensors-20-04171-f002:**
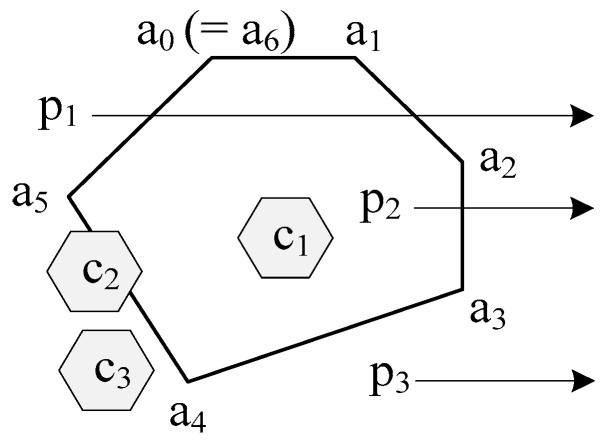
Example of an area, points, and cells. This area is a polygon composed by points a_0_, a_1_, …, and a_5_. To this area, point p_1_ and point p_3_ are not-inside and point p_2_ is inside, both cells c_1_ and c_2_ are overlapped and c_3_ is not overlapped. Moreover, cell c_1_ is also termed as inside this area.

**Figure 3 sensors-20-04171-f003:**
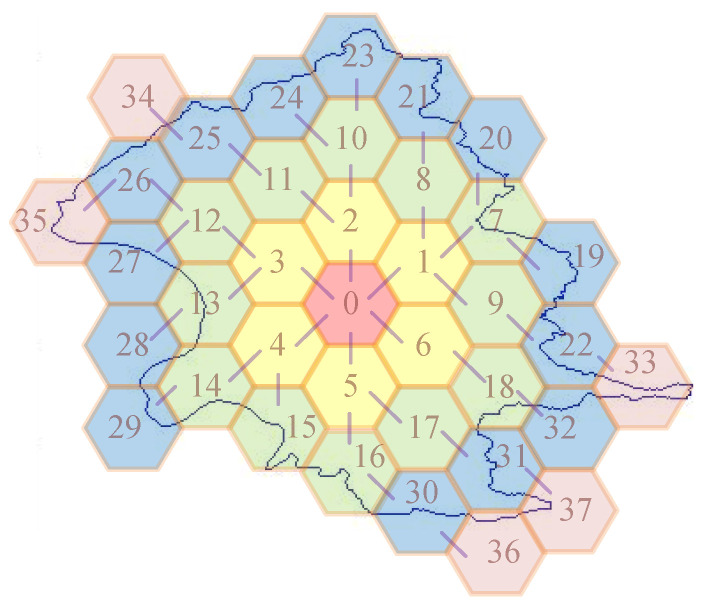
Example of Cell Allocation to an area. The allocation is started from *c*_0_. In each iteration, some cells will be allocated. After five iterations, the allocation is finished.

**Figure 4 sensors-20-04171-f004:**
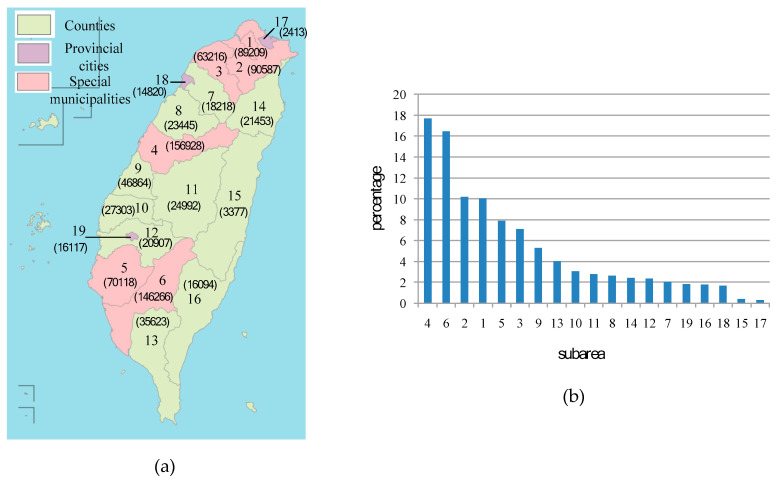
The experimental area with 887,950 points. This area currently contains 6 special municipalities, 10 counties and 3 provincial cities. These subareas are numbered as 1–19: (**a**) The number of points in each subarea. (**b**) The point ratio of each subarea to 16 subareas.

**Figure 5 sensors-20-04171-f005:**
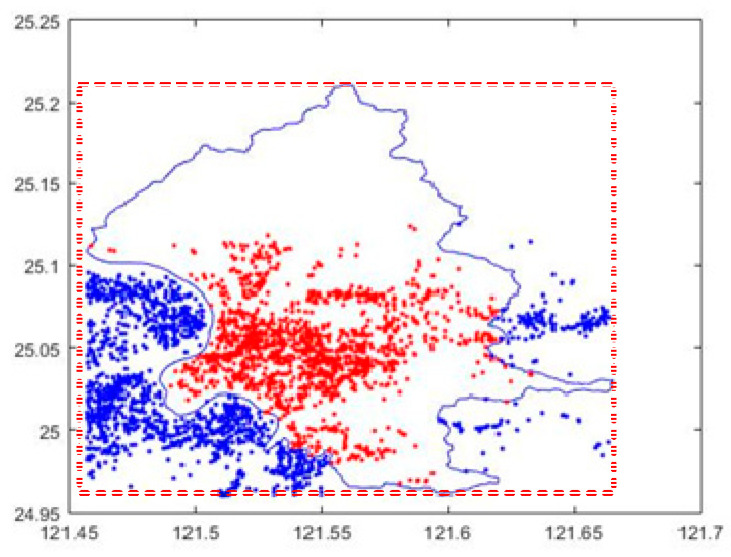
Point distribution of subarea 1 and its neighborhood. The range is longitude 121.45 to 121.67 and latitude 24.95 to 25.22. The number of points in this range is 93,549. Randomly, 1600 points inside subarea 1 and 1600 points not-inside subarea 1 are marked on this range.

**Figure 6 sensors-20-04171-f006:**
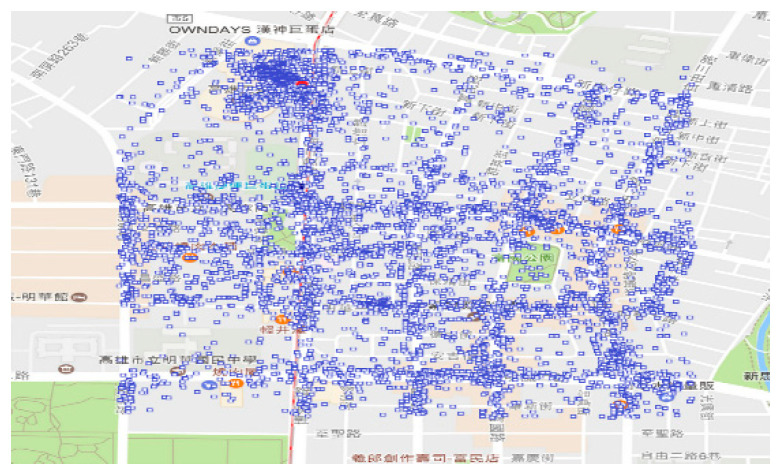
Distribution of points in hot spot one with 6024 points.

**Figure 7 sensors-20-04171-f007:**
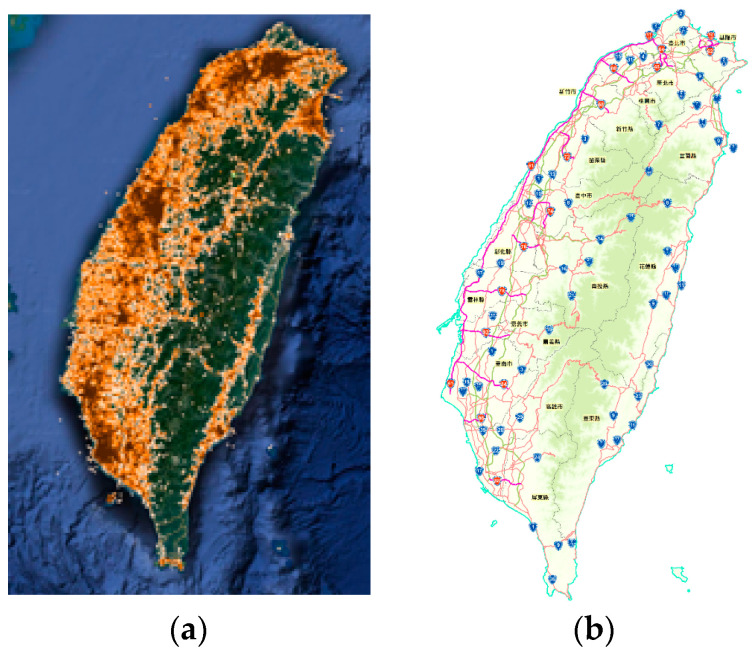
(**a**) Distribution of spots and (**b**) distribution of highways. The layout of (**a**), respectively, coincides with the layout of (**b**) and the layout of Figure 4a.

**Table 1 sensors-20-04171-t001:** The vertices of cell *c*. For a cell *c* with radius *r*, the coordinates of each vertex are as follows.

Vertex	Coordinate
*c*_V0_ (=*c*_V__6_)	(*x*(*c*) + *r*, *y*(*c*))
*c* _V1_	(*x*(*c*) + 0.5*r*, *y*(*c*) + 0.5$\sqrt{3}$*r*)
*c* _V2_	(*x*(*c*) − 0.5*r*, *y*(*c*)+ 0.5$\sqrt{3}$*r*)
*c* _V3_	(*x*(*c*) − *r*, *y*(*c*))
*c* _V4_	(*x*(*c*) − 0.5*r*, *y*(*c*) − 0.5$\sqrt{3}$*r*)
*c* _V5_	(*x*(*c*) + 0.5*r*, *y*(*c*) − 0.5$\sqrt{3}$*r*)

**Table 2 sensors-20-04171-t002:** The neighbors of cell *c*. For a cell *c* with radius *r*, the coordinates of each neighbor are as follows.

Neighbor	Coordinate
*c*_N0_ (=*c*_N6_)	(*x*(*c*) + 1.5*r*, *y*(*c*) + 0.5$\sqrt{3}$*r*)
*c* _N1_	(*x*(*c*), *y*(*c*) + *r*)
*c* _N2_	(*x*(*c*) − 1.5*r*, *y*(*c*) − $\sqrt{3}$*r*)
*c* _N3_	(*x*(*c*) − 1.5*r*, *y*(*c*) − $\sqrt{3}$*r*)
*c* _N4_	(*x*(*c*), *y*(*c*) + −$\sqrt{3}$*r*)
*c* _N5_	(*x*(*c*) + 1.5*r*, *y*(*c*) − 0.5$\sqrt{3}$*r*)

**Table 3 sensors-20-04171-t003:** Population density and space density based on number of points.

Subarea	Population	Space (km^2^)	No. of Points	Population Density (#/Population × 100%)	Space Density (#/Space)
1	2,687,629	271.8	89,209	3.32	328.22
2	3,984,051	2052.57	90,587	2.27	44.13
3	2,171,127	1220.95	63,216	2.91	51.78
4	2,778,182	2214.90	156,928	5.65	70.85
5	1,886,267	2191.65	70,118	3.72	31.99
6	2,777,873	2951.85	146,266	5.27	49.55

#: number of points.

**Table 4 sensors-20-04171-t004:** Spot statistics with points.

No. of Points	Spots	Percentage
less than 1000	15,347	98.55
1000–2000	152	0.98
2000–3000	49	0.31
3000–4000	13	0.08
4000–5000	4	0.03
5000–6000	7	0.04
6000–7000	1	0.01
Total	15,573	100

**Table 5 sensors-20-04171-t005:** The first 12 hot spots.

Hot Spot	Location	Coordinate	No. of Points
1	subarea 6	120.3, 22.67	6024
2	subarea 6	120.3, 22.62	5731
3	subarea 4	120.64, 24.18	5527
4	subarea 6	120.3, 22.63	5506
5	subarea 4	120.66, 24.16	5428
6	subarea 4	120.68, 24.16	5154
7	subarea 4	120.68, 24.15	5152
8	subarea 6	120.3, 22.64	5043
9	subarea 4	120.65, 24.16	4855
10	subarea 19	120.44, 23.48	4544
11	subarea 4	120.68, 24.14	4424
12	subarea 9	120.54, 24.08	4056

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
