# Peer review of "A GNSS-Based Crowd-Sensing Strategy for Specific Geographical Areas"

_sensors, 2020, doi:10.3390/s20154171_

Round 1

Reviewer 2 Report

The paper discusses the crowd-sourced positioning of all smartphone users within specific areas or detection if the user is in such an area. One of the claimed application areas is epidemics management.

It is a bit confusing that 5G technology is advocated in the introduction, but the study uses GNSS / GPS positions. Also the claimed accuracy of 30 cm is typically not achieved by commodity smartphones yet.

The approach itself may be valid and useful. The scaling of the cells seems to be location specific (latitude, longitude values), right?

The privacy constraints should also be addressed in the paper and in particular in any application using the proposed strategy.

There are a lot of typos and grammatical errors in the paper, examples follow. The language needs to be checked carefully!

In the title: Crowed Sensing Strategy --> Crowd-Sensing Strategy

In abstract: locate the location --> acquire the location

crowd-sensing sensing strategy --> crowd-sensing strategy

specific geographical areas that can be used to calculate how many people are in specific geographical areas or whether the person is in a specific area. (a lot of repetition)

the data records containing latitude and longitude in a well-known social networking service platform is used --> ...are used

In introduction: integrate a multitude of sensors based on both, cellular signals, global navigation satellite system (GNSS) signals, and 3GPP independent techniques, into a hybrid positioning scheme --> integrate into a hybrid positioning scheme a multitude of sensors based on both cellular...

These systems allows --> ...allow

pinpointing to within 30 centimeters or 11.8 inches (is a fraction of inch necessary for the accuracy? 12 inches?)

The number of visits and the average stay time exceeds 15 minutes. (the number of visits is not measured in minutes)

GNSS with 5G integrated for positioning a promising research topic --> ...is a promising research topic

an specific geographical areas --> specific geographical areas

In section 2: Here the cellular architecture, that the use of regular hexagonal configuration instead of the circular configuration, can be applied --> Here the cellular architecture using a regular hexagonal configuration instead of the circular configuration can be applied

cells cN1, cN2,…, and cN5 are used to represent the 6 neighbors of cell c (only five neighbors listed...)

of each vertex Table 2 provides --> of each vertex. Table 2 provides

are shown in Tab. (???)

outwardly (in many places) --> outwards ???

The expansion outwardly is doing until area A is completely covered by cells --> The expansion outwards is done until...

this algorithm is stop when --> this algorithm is stopped when

Crowed Sensing algorithm --> Crowd-Sensing algorithm

Figure 9. Crowed Sensing

3. Demonstration --> 4. Demonstration

The acquisition of points wan on January 2017 --> ???

the value in brackets is the number of points of the subarea --> the value in parenthesis is ...

Because the position accuracy of GNSS is already within a distance of 30 cm, by our strategy, It can positioned --> ...by our strategy, the target can be positioned

the population and the space can take in to consideration --> ...can be taken into consideration

the calculation of the targets in what spots had its utility --> ???

4. Discussion --> 5. Discussion

Round 2

Reviewer 2 Report

Please change all occurrences of "crowed-sensing" to "crowd-sensing", including in the paper title and Fig.9 capture.

Otherwise the paper starts to be in good condition, but please check carefully the language...

Round 3

Reviewer 1 Report

All comments are resolved.